# Gene Detection and Enzymatic Activity of Psychrotrophic *Bacillus cereus s.s.* Isolated from Milking Environments, Dairies, Milk, and Dairy Products

**DOI:** 10.3390/microorganisms13040889

**Published:** 2025-04-12

**Authors:** Carlos E. G. Aguilar, Gabriel Augusto Marques Rossi, Higor O. Silva, Luisa Maria F. S. Oliveira, Alenia Naliato Vasconcellos, Danielle de Cássia Martins Fonseca, Andréia Cristina Nakashima Vaz, Bruna Maria Salotti de Souza, Ana Maria Centola Vidal

**Affiliations:** 1Secretariat of Agriculture and Supply of the State of São Paulo, São Paulo 04014-900, SP, Brazil; kadugamero@hotmail.com; 2Department of Veterinary Medicine, University of Vila Velha (UVV), Vila Velha 29102-920, ES, Brazil; 3Mário Palmério University Center (UNIFUCAMP), Monte Carmelo 38500-000, MG, Brazil; higorvet@yahoo.com.br; 4College of Animal Science and Food Engineering, University of São Paulo (FZEA/USP), Pirassununga 13635-900, SP, Brazil; luisamariafso@usp.br (L.M.F.S.O.); dmartinsfonseca@usp.br (D.d.C.M.F.); acnvaz@usp.br (A.C.N.V.); anavidal@usp.br (A.M.C.V.); 5Department of Technology and Inspection of Animal Products, Veterinary School, Federal University of Minas Gerais (UFMG), Belo Horizonte 31270-901, MG, Brazil; brunasalotti@gmail.com

**Keywords:** dairy, lipolysis, microorganisms, production, proteolysis

## Abstract

*Bacillus cereus* is a spore-forming, Gram-positive bacterium that causes foodborne illness and dairy spoilage. This study analyzed *B. cereus s.s.* isolates from milking environments, raw milk, and dairy products to assess their genotypic and phenotypic traits. From 466 samples, 61 isolates were obtained: 27 from milking environments, 9 from dairy environments, 8 from raw milk, and 17 from dairy products. Genomic sequencing identified genes encoding proteolytic (BC5350, BC0666, BC2984, BC0598, BC5351, BC3383, BC2735), lipolytic (BC4862, BC2141, BC1027, BC4123, BC4345, BC5402, BC5401), and esterase (BC1954, BC4515, BC3413, BC3606) enzymes. Plate assays confirmed enzymatic activities. Proteolytic genes were more prevalent in environmental samples, followed by raw milk and dairy products. Lipolytic genes were most frequent in raw milk, followed by environmental samples and dairy products. Esterase genes were most common in dairy environments. These findings suggest that dairy processing influences the enzymatic profile of *B. cereus s.s.*, potentially impacting food safety and quality in the dairy industry. Understanding the distribution of these genes may help develop strategies to mitigate spoilage and contamination risks in dairy products.

## 1. Introduction

*Bacillus cereus* is a well-known foodborne pathogen. It is a facultative anaerobic, spore-forming microorganism with psychrotrophic behavior, typically associated with the deterioration of dairy products [1]. The spoilage occurs due to extra- or intracellular enzymes, which can be responsible for the reduced stability of dairy products. The presence of *Bacillus* sp. spores is typically observed in raw milk and represents a challenge to the dairy industry, as they are heat-resistant, remaining in the milk after pasteurization and ultra-high-temperature (UHT) treatment [2,3].

Food poisoning outbreaks caused by this opportunistic pathogen have been widely reported [4,5], and the actual number of outbreaks attributable to *B. cereus* may be higher, as detailed diagnostics to identify pathogenic strains are not widely used in microbiology laboratories [6]. Between 2000 and 2015, Brazil reported 10,666 foodborne disease outbreaks, affecting 209,240 individuals and resulting in a 0.05% mortality rate. According to the Department of Health Surveillance of the Brazilian Ministry of Health, *Bacillus cereus* was the fourth most common causative agent, accounting for 3.1% of the reported cases [7]. Additionally, studies indicate that the molecular profile of different *B. cereus* strains associated with food poisoning outbreaks in Brazil aligns with their ability to produce toxins, as well as enzymes that compromise product quality [8,9].

The alteration of flavor and texture in dairy products has been attributed to the activity of thermally stable microbial protease, responsible for off-flavors and gel formation in milk subjected to UHT treatment [10,11]. Most psychrotrophic bacteria have the ability to synthesize thermally stable proteolytic and lipolytic enzymes after sporulation, which break down the main components of milk [1].

There are three bacterial extracellular proteases, alkaline metalloprotease (Apr), neutral metalloprotease (Npr), and serine protease (Sub). Apr has broad specificity, a pH optimum of 7 to 9, and is produced by bacteria of the genera *Pseudomonas* and *Serratia*. Npr has a preference for hydrophobic substrates or large amino acid residues, with a pH optimum of 7.0. Npr has been observed in a wide range of taxa, including the genus *Bacillus* in bacteria and *Aspergillus* in fungi. Sub is a representative subtilisin produced by *Bacillus subtilis* [12].

The presence of neutral metallopeptidases stands out as the main protease produced by *Bacillus cereus*. These enzymes are encoded by the *npr* gene, which is highly conserved in *B. cereus* and is commonly considered a reference for molecular detection of proteolytic activity and estimation of the spoilage potential of this species. Their activity is influenced by environmental factors such as temperature [13,14].

The expression of several enterotoxins, hemolysins, phospholipases, and proteases is controlled by *PlcR*, the transcriptional regulator (Phospholipase C Regulator), one of the main regulators of the virulence genes of *B. cereus* [15,16,17]. *PlcR* requires *PapR*, a peptide expressed as a propeptide under the control of *PlcR*, to be active [18,19]. *PlcR* integrates at least two classes of signals: cell growth state and self-cell density via *PapR* [20,21].

Lipolysis can occur due to enzymatic activity or spontaneously during milk processing [22]. Several microorganisms produce lipases capable of hydrolyzing milk fat, some of which survive pasteurization [14,23,24]. Product defects are detected when the concentration of microorganisms reaches 5.0 to 7.0 log CFU/mL [25,26].

Studies indicate that the presence of lipases produced by psychrotrophic microorganisms can promote fat hydrolysis with the release of fatty acids and the formation of hexanoic, octanoic, decanoic, and butyric acids, which result in undesirable flavor changes in the product [27,28,29]. The main factor controlling the expression of lipase activity is the presence of lipid sources; however, lipase production is also influenced by other carbon sources, such as sugars. Additionally, lipase synthesis is affected by factors such as temperature, pH, the presence of inorganic salts, agitation, and oxygen concentration [30].

Esterases represent a diverse group of hydrolases, enzymes that catalyze the hydrolysis of a wide range of aliphatic and aromatic esters, and they are generally restricted to short-chain esters. Esterases can be synthesized by microorganisms from the genera *Pseudomonas*, *Bacillus*, *Lactobacillus*, *Rhodococcus*, *Aspergillus*, and *Geotrichum* [31]. Esterases have been studied for their potential in several industrial processes, such as the synthesis of aromas, drugs, and biopolymers, as well as separation of racemic mixtures [32,33,34]. Esterases follow the classical Michaelis–Menten kinetics for their activity to occur [35].

The understanding of the processes related to the spoilage capacity of *Bacillus cereus* through the phenotypic and genotypic characterization of proteolytic and lipolytic activities is still not clearly defined. Thus, the objective of this study was to understand the presence of genes related to the production of spoilage enzymes by *Bacillus cereus s.s.* along the dairy production chain. To achieve this, *B. cereus s.s.* isolates from milking environments, raw milk, and Brazilian dairy products were analyzed to detect the presence of these genes (genotypic characterization) and their respective enzymatic activities (phenotypic characterization), assessing their genotypic and phenotypic traits.

## 2. Materials and Methods

### 2.1. Experimental Design

The experimental design (Figure 1) of this study involved the collection and characterization of relevant samples. Samples were obtained from various environments within the dairy production chain, including milking environments on dairy farms, dairy processing facilities, and commercially available dairy products. The target microorganism, *Bacillus cereus s.s.*, was isolated in the laboratory. Samples confirmed as *B. cereus s.s.* were then subjected to genotyping to identify genes associated with proteolytic and lipolytic activities and the presence of esterase enzymes. This approach allowed for statistical analyses to assess the presence of these genes across different stages of the dairy production chain.

### 2.2. Sample Characterization

A total of 466 samples were collected from the dairy production chain in Brazil and categorized into three main sources: Dairy farms (*n* = 331): Samples were collected from 26 different environments, including bulk storage tanks (7), a manual milking pail (1), milk cans (5), milk pipelines (4), raw milk (6), milking clusters (5), and other utensils or equipment (10) in direct contact with milk; Dairy processing plants (n = 58): Collected from two dairy processing facilities in the same region, including milk trucks unloading raw milk (2), equipment surfaces (3), milk pipelines (1), and utensils (3) used in dairy production; Retail dairy products (n = 77): Obtained from commercially available dairy products such as Minas cheese (1), UHT dairy beverage (1), cheese-based cream (1), instant cappuccino (1), and Brazilian creamy cheese “requeijão” (18).

Among these, 69 *Bacillus cereus s.s.* isolates were phenotypically confirmed and categorized based on their source: equipment and utensils, raw milk and bulk storage tanks, and dairy products. These isolates were further classified into predefined phylogenetic groups. Specifically, 61 isolates carried genes responsible for the synthesis of proteolytic, lipolytic, and esterase enzymes, distributed as follows: 27 from milking environment surfaces, 8 from raw milk, 9 from dairy processing equipment and utensils, and 17 from dairy products. The methods for sample collection, biochemical identification, DNA extraction, and genetic sequencing were previously described [9].

### 2.3. Genotypic Characterization

Sequence data from 61 isolates obtained from Brazilian dairy production are archived in the NCBI GenBank repository and the Short Read Archive (SRA), associated with BioProject PRJNA390851 (Table S1, Supporting Information) [9]. Assembled genomes are also available on FigShare (10.6084/m9.figshare.5120020). To investigate the presence of genes responsible for the synthesis of proteolytic enzymes, seven genes were examined: BC5350, related to PlcR (a pleiotropic regulator affecting the npr gene); BC0666, related to inhA2 (responsible for metalloprotease enzymatic activity); BC2984, related to inhA3 (responsible for metalloprotease enzymatic activity); BC0598, related to npr (responsible for proteolytic activity); BC5351, related to nprB (responsible for neutral metalloprotease enzymatic activity); BC3383, related to nprC (responsible for neutral metalloprotease enzymatic activity); and BC2735, related to nprP2 (responsible for neutral metalloprotease enzymatic activity) [36,37,38].

To investigate the presence of genes responsible for the synthesis of lipolytic enzymes, seven lipase genes were analyzed: BC4862 (Lipase), BC2141 (Lipase), BC1027 (Lipase), BC4123 (Lipase/Acylhydrolase with a GDSL-like motif), BC4345 (Lipase), BC5402 (Transcriptional regulator, LacI family), and BC5401 (Lipase/Acylhydrolase with a GDSL-like motif). To verify the presence of genes responsible for the synthesis of esterase enzymes, four esterase genes were examined: BC1954 (Esterase), BC4515 (Esterase), BC3413 (Esterase), and BC3606 (Esterase) [38].

### 2.4. Phenotypic Characterization

For the evaluation of proteolytic activity of *Bacillus cereus s.s.*, successive decimal dilutions were performed up to 10^−5^. Then, 0.1 mL was inoculated into milk agar (standard agar plus 1% skim milk powder), followed by incubation at 28 °C for 24–48 h [39]. A positive result was evidenced by the appearance of a clear halo around the colony due to the degradation of casein present in the milk, which was then classified as positive proteolytic activity.

In order to evaluate the lipolytic activity, after dilution, the isolates were plated on tributyrin agar, prepared with standard agar for counting added with 1% tributyrin [39]. The plates were incubated at 28 °C for 5 days. A positive result was evidenced by the appearance of colonies forming a transparent halo, which was then classified as positive lipolytic activity.

### 2.5. Statistical Analysis

The total number of expressed genes was counted using the “COUNT IF” function in Microsoft Office Excel (2016). To assess the association between qualitative variables, non-parametric frequency statistics were applied. The significance test used was the Chi-Square, with a significance level of 5%. Data analysis was performed with the help of the FREQ procedure of the Statistical Analysis System software, version 9.4 (SAS, Cary, NC, USA, 2018).

## 3. Results

### Presence of Genes in Bacillus cereus s.s. Isolates

In the 61 isolates of *Bacillus cereus s.s.*, the presence of several genes involved in proteolytic activity (BC5350, BC0598, BC0666, BC2984, BC5351, BC3383, and BC2735) was identified (Table 1). These genes are responsible for the synthesis and regulation of enzymes that degrade proteins, such as metalloproteinases and neutral proteases. For the variables npR (BC0598), corresponding to the pleiotropic regulator gene, and nprB (BC5351) statistically significant differences were observed (*p*-value = 0.043 and 0.031, respectively). The highest frequency and percentage of positive activity were found in the milking environment compared to the dairy processing, raw milk, and dairy product environments.

In all isolates from the milking environment, 27 (100%) of the samples showed the presence of genes associated with the pleiotropic regulation of the npr gene (PlcR_BC5350), metalloproteinase activity (inhA2_BC0666), and regulation of proteolytic activity (npr_BC0598). In isolates obtained from raw milk, genes responsible for metalloproteinase activity (inhA2_BC0666 and inhA3_BC2984) and proteolytic regulation (npr_BC0598) were also consistently found. Similarly, isolates from dairy environments displayed genes for pleiotropic regulation (PlcR_BC5350), metalloproteinase activity (inhA2_BC0666 and inhA3_BC2984), and proteolytic activity (npr_BC0598). However, in isolates from dairy products, the genes responsible for pleiotropic regulation (PlcR_BC5350) and metalloproteinase activity (inhA2_BC0666) were detected in 16 of 17 samples (Table 1). A 100% (9) frequency of proteolytic activity was observed only in isolates from dairy environments, followed by milking environments (96.29%—26), raw milk (87.5%—7), and dairy products (82.35%—14).

In the same 61 isolates, genes responsible for lipolytic enzyme synthesis (BC4862, BC2141, BC1027, BC4123, BC4345, BC5402, and BC5401) were also detected (Table 2). Lipases, which are crucial for the breakdown of fats, were regulated by these genes. In 100% (27) of the isolates from the milking environment, the presence of genes BC4862 and BC5401 was observed. In raw milk isolates, 100% (8) of the lipase regulator genes (BC4862, BC1027, BC4123, BC5402, and BC5401) were present. Isolates from dairy environments exhibited the same genes (BC4862, BC2141, BC1027, BC4123, BC4345, and BC5402) in all samples. In dairy product isolates, however, only the lipase regulator gene BC5402 was present in 100% (17) of the isolates (Table 2). Lipolytic activity was observed in 50% (4) of raw milk isolates, followed by 33.33% (9, 3) of the milking and dairy environment isolates, and only 29.41% (5) of the dairy product isolates. There was no statistically significant difference between the frequency of genes and lipolytic activity from different sampled locations.

The presence of genes responsible for esterase enzyme synthesis (BC1954, BC4515, BC3413, and BC3606) was also confirmed in all isolates (Table 3). Esterases are involved in the hydrolysis of ester bonds, and their regulation is crucial in lipid metabolism. In milking environment isolates, the esterase regulator gene (BC3606) was present in all samples. In raw milk isolates, esterase regulator genes BC4515 and BC3606 were detected in all samples, while in dairy environments, BC4515, BC3413, and BC3606 were consistently present. In dairy product isolates, the esterase regulator genes (BC4515, BC3413, and BC3606) were detected in all samples (Table 3). There was no statistically significant difference between the frequency of genes from different sampled locations.

Upon analyzing the isolates from different locations, the proteolytic activity was frequent in all sites, except for dairy derivatives, showing lower proteolytic activity in the latter. A similar pattern was observed in lipolytic activity, with higher frequencies in most locations except for dairy products, suggesting that industrial processing may affect the expression of the related genes. As for esterase activity, lower frequencies were observed, especially in raw milk and dairy products, compared to the milking environment, too. Thus, it can be inferred that the presence of the spoilage genes is more prevalent in the beginning of the dairy chain.

## 4. Discussion

Corroborating the findings where the prevalence of *Bacillus cereus s.s.* and enzymatic activity in dairy products were evaluated, it was observed that 13% of the analyzed samples tested positive for *B. cereus s.s.* [9]. Among these, over 91.8% exhibited proteolytic activity, while less than 34.4% showed lipolytic activity. Proteases were released across all temperature ranges evaluated [40]. In a similar study analyzing dairy product samples, 100% of cream samples were found to contain *B. cereus*, with 64% showing proteolytic activity and 16% displaying lipolytic activity. In yogurt samples, the presence of *B. cereus* was also detected in all samples, with 84% exhibiting proteolytic activity and 20% showing lipolytic activity [41].

Maziero et al. [42] evaluated the relationship between the incidence and lipolytic and proteolytic activity of *B. cereus* in UHT milk and observed that 16.36% of the samples were identified as strains of this microorganism, of which 100% exhibited proteolytic activity and none exhibited lipolytic activity. After 96 h of incubation at 30 °C, the authors reported that 55.6% of the isolates displayed lipolytic activity. The proteolysis of UHT milk during storage at ambient temperature is a primary limitation of its shelf life due to textural changes such as increased viscosity, which in some cases leads to gel formation [43].

On the other hand, data in the literature on the spoilage potential of spore-forming bacteria in refrigerated raw milk showed that *Bacillus* spp. were detected in 55% of the samples, with 72.7% exhibiting both proteolytic and lipolytic activities, 9.1% showing only proteolytic activity, and 18.2% showing only lipolytic activity, with the majority of isolates demonstrating both activities [3]. Furthermore, in a study investigating milk contamination during milking by proteolytic and lipolytic microorganisms, 55.17% of the analyzed samples tested positive for the presence of *B. cereus*, of which 20% exhibited only proteolytic activity and 44.14% showed lipolytic activity [44].

In the present study, the presence of genes responsible for the synthesis of proteolytic enzymes was confirmed. Among the *B. cereus s.s.* strains isolated from the milking environment, dairy environment, raw milk, and dairy products, a frequency of 100% was observed for the regulators PlcR and npr, along with high levels of proteolytic activity in these environments. This indicates that, among all the locations analyzed, the most intense proteolytic activity was due to the maximum frequency of the gene and its regulator.

A study analyzing biochemical and genotypic characteristics of *Bacillus cereus* and *Bacillus thuringiensis* isolated from cheese samples observed that most strains produced extracellular enzymes in dairy product samples [45]. Similar results were observed by Montanhini et al. (2013) [14] in strains of *B. cereus* isolated from powdered milk, UHT milk, and pasteurized milk. These researchers documented the presence of the *npr* gene and proteolytic activity.

In general, the occurrence of *B. cereus s.s.* in all the products analyzed can be attributed to poor raw material quality or unsatisfactory hygienic–sanitary practices, as well as its ability to adhere and form biofilms. This may facilitate contamination of dairy products throughout the production chain. Moreover, ideal refrigeration conditions inhibit the growth of microorganisms and the production of enzymes, making storage temperature a key factor in maintaining quality [46].

According to the previous findings, it can be inferred that among the spoilage activities analyzed, refrigerated raw milk is more vulnerable to protein degradation. This can compromise the concentration of dairy components and the industrial yield of future derivatives. Therefore, it is important to emphasize that preventing contamination is essential to extend shelf life, as contamination by spore-forming microorganisms can lead to enzymatic degradation of dairy products [47].

The occurrence of these spoilage microorganisms, resulting from inadequate hygiene practices during milking and processing, creates an environment conducive to bacterial enzyme production and interaction with available substrates. This can occur even in the presence of food safety and quality controls. For instance, a significant delay between thorough cleaning of equipment and utensils, or the presence of residual water in tanks, can contribute to contamination [48,49].

In this study, the presence of the extracellular GDSL-like lipase/acylhydrolase enzyme (BC4123 and BC5401) was identified. Upon analyzing isolates obtained from different locations, the highest frequency of lipase and GDSL-type lipase/acylhydrolase genes was found in dairy environments, followed by raw milk, the milking environment, and dairy products. However, lipolytic activity was highest in isolates from raw milk, followed by those from the milking and dairy environments, with the lowest activity observed in dairy products.

Teh et al. [50] noted that lipases produced by bacteria originating from raw milk can remain active after heat treatment, causing lipolysis, which can reduce the quality and shelf life of dairy products. These thermostable lipases can lead to rancidity as a result of the hydrolysis of milk fat into free fatty acids. In addition to sensory defects, lipolysis can alter the physicochemical properties of milk, leading to instability in dairy beverages.

Studies by Deeth [51] and Wiking [52] indicate that lipolysis is less likely to occur in pasteurized milk due to the low storage temperature and relatively short shelf life. A similar phenomenon was observed in the present study, where the frequencies of genes responsible for the synthesis of lipolytic enzymes and lipolytic activity of *B. cereus* s.s. were less frequently identified in dairy derivatives. This reinforces the assertion that lipolysis typically occurs only at the end of pasteurized milk’s shelf life and is characterized by the appearance of undesirable flavors related to the high concentration of free fatty acids and rancidity, one of the most problematic defects in butter, for example [53].

According to Teng et al. [54], free fatty acids are formed as a result of lipase action on milk triacylglycerols, particularly short- and medium-chain fatty acids. Although they are associated with the spoilage of various dairy products, microbial lipases are also used in the dairy industry for several purposes. These include improving and enhancing the flavor of yogurt and cheese, butter, and cream lipolysis according to processing needs, and even accelerating cheese ripening [55].

In addition to endogenous lipase, other enzymes of microbial origin may also be present in dairy products. These enzymes are primarily extracellular lipases produced by psychrotrophic bacteria. They typically exhibit an alkaline pH and have optimal activity at temperatures ranging from 40 to 50 °C. Moreover, these lipases are thermally stable, with some demonstrating resistance even to UHT treatments [56].

According to Santos et al. [55], for this reason, lipases are commonly present in UHT milk, which can impact the product’s shelf life and quality. In a study conducted by Andrewes [57], lipase was detected in UHT milk through the formation of methyl esters. This study found that the greater the amount of lipases in the sample, the more pronounced the product’s deterioration, leading to the development of off-flavor, like rancid-tasting, within two weeks. According to Mordor Intelligence [58], lipases account for 10% of the global enzyme market, and this market is expected to grow by 8.8% by 2025. Enzymes capable of hydrolyzing only acylglycerols with fewer than 10 carbons in their chain are generally referred to as esterases, differing from lipases in their substrate specificity [59,60,61].

In powdered milk, bacterial lipases may retain residual activity, potentially affecting product quality over time [55]. Deeth [51] observed that these enzymes act slowly during storage, with lipolytic activity becoming detectable even months after production. As a result, their persistence can lead to economic losses, especially when powdered milk is used as an ingredient in the food industry.

Upon analyzing the isolates obtained, a higher frequency of genes responsible for esterase synthesis was observed in isolates from dairy environments, followed by raw milk, the milking environment, and with the lowest frequency in dairy products. The classification of esterases is difficult due to the lack of a universally accepted classification system. The most commonly used system is likely the one proposed by Holmes and Masters [62], which primarily differentiates enzymes based on their specific inhibition by various agents rather than substrate specificity.

Esterases have been investigated for their potential applications in various industrial processes, including flavor synthesis, pharmaceuticals, biopolymers, and the separation of racemic mixtures [33,34]. These enzymes can be sourced from a diverse range of microorganisms, such as *Pseudomonas* sp., *Bacillus* sp., *Lactobacillus* sp., *Rhodococcus* sp., *Aspergillus* sp., and *Geotrichum* sp. [31]. Their broad enzymatic capabilities make them valuable for biotechnological and industrial applications.

Older studies by Forster et al. [63] and Marquardt and Forster [64] observed high activity of arylesterase and carboxylesterase enzymes in abnormal milks, such as colostrum and mastitic milk. Purr et al. [65] reported the presence of esterase activity with considerable thermal stability, inactivated in milk at 88 °C. According to Olivecrona et al. [66], the total esterase activity in normal milk can be estimated at around 0.05 mmol ml^−1^ min^−1^. This may not be the case for some abnormal milks, where esterase levels are significantly elevated (10–12 times higher, and up to 37 times higher). The significance of these esterases in milk and their relationships with each other and with esterases from other tissues remain to be determined.

By identifying the genetic determinants of enzymatic activity, research in this area can contribute to the development of control strategies to mitigate microbial spoilage and improve the safety of dairy products. The prevalence of proteolytic and lipolytic enzymes in the dairy chain could be reduced through stringent hygiene measures in dairy environments, raw milk, and dairy products. Additionally, the presence of *Bacillus cereus s.s.* esterase genes may influence spoilage and product stability. Therefore, implementing effective control measures is essential to ensuring food safety and maintaining dairy product quality, given the potential risks associated with the enzymatic activity of this microorganism.

## 5. Conclusions

In this study, it was observed that the frequency of genes related to the enzymatic activity of proteases, lipases, and esterases was higher in *Bacillus cereus s.s.* isolated from dairy environments, followed by isolates from raw milk and milking environments. The lowest expression was observed in isolates from dairy products. Based on these results, it can be inferred that the processes applied during the stages of dairy product production may interfere in a way that leads to changes in the genomic activity associated with the expression of specific enzymes and consequently in the enzymatic activity expressed by the microorganism. Further research is needed to corroborate this hypothesis and assess which specific stage could have such an effect.

## Figures and Tables

**Figure 1 microorganisms-13-00889-f001:**
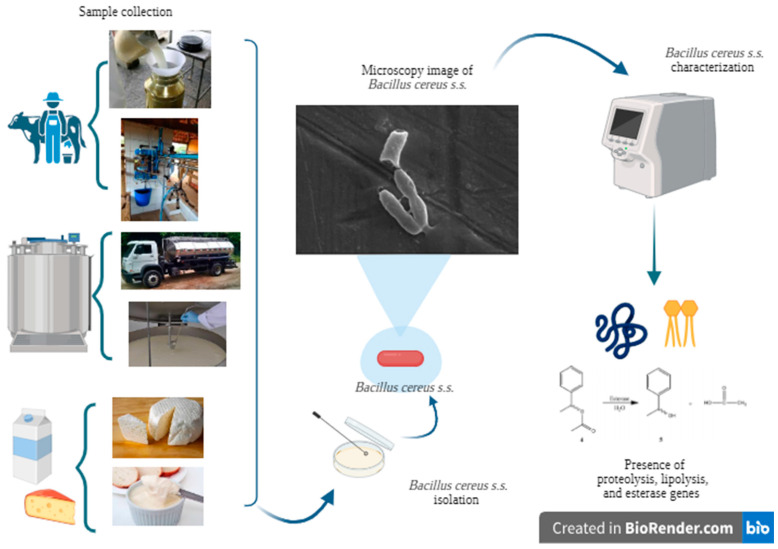
Experimental design for the selection of isolates from sample collection, isolation process, and characterization of *B. cereus s.s.* to the identification of target genes. Figure created in BioRender^®^ (free version) by authors (2025).

**Table 1 microorganisms-13-00889-t001:** Contingency table of the frequency and percentage of genes responsible for the proteolytic enzyme synthesis and proteolytic activity of *Bacillus cereus s.s.* isolated from different sampled locations.

Genes	M.E.		D.E.		R.M.		D.P.		Chi-Square Test			
	Freq ^1^	% ^2^	Freq ^1^	% ^2^	Freq ^1^	% ^2^	Freq ^1^	% ^2^	N	DF	x^2^	*p*_Value
PlcR (BC5350)	27	44.26	9	14.75	7	11.48	16	11.48	61	3	3.73	0.292
npR (BC0598)	27	44.26	9	14.75	8	13.11	14	22.95			8.17	0.043 *
inhA2 (BC0666)	27	44.26	9	14.75	8	13.11	16	26.23			2.63	0.452
inhA3 (BC2984)	26	42.62	9	14.75	8	13.11	15	24.59			2.67	0.446
nprB (BC5351)	14	22.95	4	6.56	5	8.20	2	3.28			8.90	0.031 *
nprC (BC3383)	20	32.79	7	11.48	3	4.92	13	21.31			4.87	0.181
nprP2 (BC2735)	23	37.70	7	11.48	3	4.92	13	21.31			7.63	0.054
Prot. activity	26	42.62	9	14.75	7	11.48	14	22.95			3.74	0.291
TOTAL	27	44.26	9	14.75	8	13.11	17	27.87				

^1^ Relative frequency of positive isolates; ^2^ Absolute percentage of positive isolates. Notes: Prot. activity = Proteolytic activity, M.E. = Milking environment, D.E. = Dairy environment, R.M. = Raw milk, D.P. = Dairy products, N = Total number of observations, DF = Degrees of freedom, x^2^ = Chi-square, *p*_value = Probability of statistical significance. * Statistically significant.

**Table 2 microorganisms-13-00889-t002:** Contingency table of the frequency and percentage of the genes responsible for the lipolytic enzyme synthesis and lipolytic activity of *Bacillus cereus s*.s. isolated from different sampled locations.

Genes	M.E.		D.E.		R.M.		D.P.		Chi-Square Test			
	Freq ^1^	% ^2^	Freq ^1^	% ^2^	Freq ^1^	% ^2^	Freq ^1^	% ^2^	N	DF	x^2^	*p*_Value
BC4862	27	44.26	9	14.75	8	13.11	16	26.23	61	3	2.63	0.452
BC2141	20	32.79	9	14.75	5	8.20	13	21.31			3.78	0.287
BC1027	26	42.62	9	14.75	8	13.11	16	26.23			0.96	0.812
BC4123	24	39.34	9	14.75	8	13.11	16	26.23			2.12	0.548
BC4345	24	40.00	9	15.00	6	10.00	16	26.67			3.41	0.333
BC5402	26	42.62	9	14.75	8	13.11	17	27.87			1.28	0.734
BC5401	27	44.26	8	13.11	8	13.11	16	26.23			3.29	0.349
BC2519	7	11.67	2	3.33	2	3.33	0	0.00			5.25	0.155
BC2449	16	26.23	5	8.2	6	9.34	12	19.67			1.28	0.734
Lip. activity	9	15.52	3	5.17	4	6.9	5	8.62			1.07	0.785
TOTAL	27	44.26	9	14.75	8	13.11	17	27.87				

^1^ Relative frequency of positive isolates; ^2^ Absolute percentage of positive isolates. Notes: Lip. activity = Lipolytic activity, M.E. = Milking environment, D.E. = Dairy environment, R.M. = Raw milk, D.P. = Dairy products, N = Total number of observations, DF = Degrees of freedom, x^2^ = Chi-square, *p*_value = Probability of statistical significance.

**Table 3 microorganisms-13-00889-t003:** Contingency table of the frequency and percentage of the genes responsible for the esterase enzyme synthesis of *Bacillus cereus s.s.* isolated from different sampled locations.

Genes	M.E.		D.E.		R.M.		D.P.		Chi-Square Test			
	Freq ^1^	% ^2^	Freq ^1^	% ^2^	Freq ^1^	% ^2^	Freq ^1^	% ^2^	N	DF	x^2^	*p*_Value
BC1954	23	37.7	7	11.47	5	8.19	12	19.67	61	3	5.67	0.461
BC4515	25	40.98	9	14.75	8	13.11	16	26.22			2.25	0.896
BC3413	23	37.7	9	14.75	6	26.22	16	26.22			4.72	0.580
BC3606	27	44.26	9	14.75	8	13.11	16	26.22			2.63	0.452
TOTAL	27	44.26	9	14.75	8	13.11	17	27.87				

^1^ Relative frequency of positive isolates; ^2^ Absolute percentage of positive isolates. Notes: M.E. = Milking environment, D.E. = Dairy environment, R.M. = Raw milk, D.P. = Dairy products, N = Total number of observations, DF = Degrees of freedom, x^2^ = Chi-square, *p*_value = Probability of statistical significance.

## Data Availability

The original contributions presented in this study are included in the article. Further inquiries can be directed to the corresponding author.

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
