# Peer review of "Gene Detection and Enzymatic Activity of Psychrotrophic Bacillus cereus s.s. Isolated from Milking Environments, Dairies, Milk, and Dairy Products"

_microorganisms, 2025, doi:10.3390/microorganisms13040889_

Round 1

Reviewer 1 Report

Comments and Suggestions for Authors
  1. The main question addressed in this study was to detect the presence of genes and enzymatic activities in Bacillus cereus s.s. isolates from dairy environments, raw milk and Brazilian dairy products..
  2. The topic of the research undertaken by the authors is interesting. The authors used 61 B. cereus isolates. They assessed the presence of specific proteolytic and lipolytic genes by isolate origin. In this context, they assessed the proteolytic and lipolytic activity of bacterial strains.
  3. The manuscript contributes new knowledge to the subject compared to other published articles by identifying genes in isolates collected from the dairy environment and dairy products from a region of Brazil. The activity of these enzymes affects the quality of dairy products.
  4. Notes in the methods section:
  • Lines 89, 118: Correct spelling of Latin name Bacillus subtilis to italics.
  • Line 119: Should be 10-5.
  1. I have no comments on the research results section.
  2. The discussion is complete and properly discussed with the results of other authors.

       One comment:

  • The previous reviewer's comment should not appear in the manuscript. Lines 189-191: „Authors should discuss the results and how they can be interpreted from the perspective of previous studies and of the working hypotheses. The findings and their implications should be discussed in the broadest context possible. Future research directions may also be highlighted”
  1. The conclusion is complete. In the summary, the authors indicated the strengths and weaknesses of the conducted study.
  2. References are adequate. 61 articles were used, 46% of which are from the last 10 years and 44% from last 5 years.
  3. Tables are properly prepared and legible.

Author Response

Dear Reviewer, 

Thank you for your time and valuable comments. We have incorporated your suggestions, which have certainly improved the quality of our manuscript. Additionally, we appreciate your kind remarks.

Please see the attachment for the full text.

Best regards.

Reviewer 2 Report

Comments and Suggestions for Authors

This article provides information on the gene detection and enzymatic activity of psychrotrophic Bacillus cereus s.s. isolated from milking environments, dairies, milk, and dairy products. It is in general appropriately organized, carried out and written, however there are some points that should be corrected or clarified. 

How were the presented differences examined? Chi-square test? P-values? It is very important to explain this.

L64: "...18,8]. Product..."

L90: "Among them" instead of "Of these"

L94: Please delete "by authors"

L98-101: Tables S1 and S2 are not provided

L101: Please delete "(Rossi et al., 2018)"

L144-164: Please check font

L162: In 16 of 17 isolates or 100%?

L175: In all samples or in 16 of 17 samples?

L189-192: Please delete

L194: How was this percentage calculated?

L197: In your study or that of [35]?

L202: "Maziero et al. [37] evaluated the relationship..."

L203: "...milk, and observed..."

L210: "On the other hand, data in the literature on the spoilage..."

L226-227: "...observed by other authors [8] in..."

L235: "According to the previous findings" instead of "Given the above"

L241-244: Please rephrase

L251: "...was the highest..."

L322: "6. Patents"?

Comments on the Quality of English Language

The English could be improved to more clearly express the research

Author Response

Dear Reviewer,

Thank you for your time and valuable comments. We have incorporated your suggestions, which have significantly improved the quality of our manuscript.

Please see the attachment for the full text.

Best regards,

Reviewer 3 Report

Comments and Suggestions for Authors

Microorganisms

Manuscript Draft

Manuscript Number: 3541066

Title: Gene detection and enzymatic activity of psychrotrophic Bacillus cereus s.s. isolated from milking environments, dairies, milk, and dairy products

Article Type: Research article

General Comments on MDPI Questions that Reviewers must answer:

  • Is the manuscript clear, relevant for the field and presented in a well-structured manner? 

This manuscript is written relatively clearly, is somewhat well-structured, and is potentially relevant to the field since it focuses on improving understanding of the genotypic and phenotypic traits of Bacillus cereus which spoils dairy products resulting in food-bourne illness. Given the potential contribution of this research, this manuscript has potential but requires more improvement to warrant publication in MDPI Microorganisms. Please make the following TEN general improvements/clarifications:

1) For a research article, the length of the writing is short. Please increase the amount of writing in all sections of the manuscript.

2) In the last paragraph of the Introduction section, please clearly state the goal(s) and then the objective(s) of the research. The overall goal of the research is not stated. What is currently written is actually plural and two goals. Were there more goals?

3) Paragraphs are a minimum of 3 sentences (1 topic sentence followed by at least 2 supporting sentences). Please correct this everywhere in the manuscript (e.g., L130-132, L299-302, etc.).

4) The start of the Introduction section needs a couple of paragraphs added on the Brazilian dairy industry as well as food safety issues related to Bacillus cereus in Brazil and more generally around the world. How many illnesses and deaths annually? There needs to be better introductory context provided here.

5) Please add a Figure 1 showing what Bacillus cereus looks like. Please add a Figure 2 showing the reader what the a) milking environment, b) raw milk, and c) dairy products involved in the study look like.

6) Better “visual” presentation of the results is required as a newly added Figure 3.

7) The Materials and Methods section needs improvement. There needs to be more in-depth description of the three environments. What exactly is a milking environment? Is it a milking parlor? More description is needed. How many farms were sampled? Where were the raw milk samples taken from? Where were the dairy products sampled? What exactly were the dairy products?

8) Please make sure that the Discussion section has two subsections: 4.1. Comparisons to Prior Research and 4.2. Research Implications. What is currently written is the first sub-section. Please add the second one so that it is clear to the reader why the research results are important and discuss the bigger picture implications of the current study. It is important in applied research to answer the “So What?” question. What about the research results lends itself to improving food safety? Note that providing better introductory context in 3) above now helps the reader to understand what will be written in sub-section 4.2.

9) Please delete writing that was retained verbatim from the Word template (e.g., 189-192).

10) When referring to cited literature, please do not use a term like “Authors” on L202. Use the lead author’s name in this case like you have done elsewhere in the manuscript adding et al. if there are three or more co-authors. You can include the year in parentheses after this if it is critical for contextual understanding.

Please also make the following THREE minor edits and clarifications:

1) Please capitalize all major words in the title on L2-4

2) More detailed address information is required for all co-authors on L8-14

3) On L30, the keywords need to be in alphabetical order with the first one capitalized:

 Keywords: Bacillus cereus, dairy; lipolysis; microorganisms; production; proteolysis

  • Are the cited references mostly recent publications (within the last 5 years) and relevant? Does it include an excessive number of self-citations?

Only 8 of the 61 cited references have been published within the last 5 years since 2019. Please add more recent citations. The citations appear relevant to the research topic. There are no excessive self-citations.

  • Is the manuscript scientifically sound and is the experimental design appropriate to test the hypothesis?

The manuscript is scientifically thorough and the analyses are comprehensive. However, better description of the methods is required.

  • Are the manuscript’s results reproducible based on the details given in the methods section?

The manuscript’s results cannot be reproducible after reading the 2. Materials and Methods section since it is not clear what exactly was being sampled for example.

  • Are the figures/tables/images/schemes appropriate? Do they properly show the data? Are they easy to interpret and understand? Is the data interpreted appropriately and consistently throughout the manuscript? Please include details regarding the statistical analysis or data acquired from specific databases.

The quality of the tables are fine for the most part. Please add requested figures.

  • Are the conclusions consistent with the evidence and arguments presented?

The Conclusions are consistent with the evidence and arguments presented. Please add more detail at the end of the Conclusions on how future research can improve upon the current work.

  • Please evaluate the data availability statements to ensure it is adequate.

The Back Matter sections between the Conclusions and the References is OK.

Author Response

Dear Reviewer,

Thank you for your time and valuable comments. We have carefully incorporated your suggestions, which have significantly improved the quality of our manuscript.

We truly appreciate your constructive feedback, which has been invaluable in refining our manuscript. We hope that the revisions meet your expectations. Thank you once again for your time and thoughtful review.

Please see the attachment for the full text. 

Best regards,

Round 2

Reviewer 2 Report

Comments and Suggestions for Authors

Authors made the necessary amendments and I suggest the acceptance of their article

Author Response

Dear Reviewer,

We would like to sincerely thank you once again for your insightful observations and comments. We are confident that these refinements have further enhanced the quality of our manuscript. We appreciate your valuable feedback.

Best regards,

Reviewer 3 Report

Comments and Suggestions for Authors

Microorganisms

Manuscript Draft

Manuscript Number: 3541066

Title: Gene Detection and Enzymatic Activity of Psychrotrophic Bacillus cereus s.s. Isolated from Milking Environments, Dairies, Milk, and Dairy Products

Article Type: Research article

Please make the following minor final edits:

1) Do not indent L102.

2) After L102, please add a minimum 3-sentence paragraph describing the figure below.

3) On L107, add a period so: 2.

4) Make L168-176 into one paragraph.

5) Improve formatting for Table 1 and Table 2 such as widening the first column so all words fit on one row and making sure Freq. and p-value headers fits on one row.

6) Remove blank gap between L193-194 and L233-234.

7) On L269, change to “and pasteurized milk. These researchers documented the presence of…”

8) On L272, change to “…form biofilms. This may facilitate…”

9) On L277-278, change to “…to protein degradation. This can compromise the concentration…”

10) On L284, change to “…available substrates. This can occur even in the…”

11) On L312, change to “…several purposes. These include improving and…”

12) On L322-323, change to “…methyl esters. This study found that the greater…”

13) Merge L321-329 into one paragraph.

14) For all journal articles with volume(issue) in References, please make sure the issue number in parentheses is also in italics

Author Response

Dear Reviewer,

We would like to sincerely thank you once again for your insightful observations and comments. We have addressed all the minor corrections and added a paragraph describing the 2.1. Experimental Design (L104-112). Additionally, we have improved the tables and carefully reviewed the formatting of the references.

We are confident that these refinements have further enhanced the quality of our manuscript. We appreciate your valuable feedback and hope that our revisions meet your expectations.

Best regards,